# MALDI-TOF MS Based Typing for Rapid Screening of Multiple Antibiotic Resistance *E. coli* and Virulent Non-O157 Shiga Toxin-Producing *E. coli* Isolated from the Slaughterhouse Settings and Beef Carcasses

**DOI:** 10.3390/foods10040820

**Published:** 2021-04-10

**Authors:** Mohamed Tharwat Elabbasy, Mohamed A. Hussein, Fahad Dhafer Algahtani, Ghada I. Abd El-Rahman, Alaa Eldin Morshdy, Ibrahim A. Elkafrawy, Adeniyi A. Adeboye

**Affiliations:** 1Public Health Department, College of Public Health and Health Informatics, Ha’il University, Ha’il 2440, Saudi Arabia; dr.algahtani@gmail.com (F.D.A.); adeboye05@yahoo.co.uk (A.A.A.); 2Food Control Department, Faculty of Veterinary Medicine, Zagazig University, Zagazig 44519, Egypt; elged2010@yahoo.com (M.A.H.); mt.elabbasy@uoh.edu.sa (A.E.M.); 3Department of Clinical Pathology, Faculty of Veterinary Medicine, Zagazig University, Zagazig 44519, Egypt; gana660@gmail.com; 4Department of Psychological Sciences, Faculty of Early Childhood Education, University of Sadat City, Sadat City 32897, Egypt; imostafa6012@yahoo.com; 5Department of Public Health, University of Texas Health Sciences at Houston, Houston, TX 77030, USA

**Keywords:** multiple antibiotic resistance *E. coli*, Non-O157, slaughterhouse, MALDI-TOF MS and beef

## Abstract

Background: The emergence of multiple antibiotic resistance (MAR) *Escherichia coli* (*E. coli)* and virulent non-O157 Shiga toxin-producing *Escherichia coli* (STEC) poses a growing concern to the meat industry. Non-O157 STEC strains including O26, O45, O103, O111, O121, and O145 have been implicated in the occurrence of bloody diarrhea and hemolytic uremic syndrome in humans. This research assessed prevalence, matrix-assisted laser desorption/ionization-time of flight mass-spectrometry (MALDI-TOF MS) protein mass-spectra profiles, multidrug-resistance traits, polymerase chain reaction detection of virulence, and antibiotic-resistance genes of *E. coli* isolated from beef carcasses and slaughterhouse environments. Methods: A total of 180 convenience sponge samples were collected from two different sources-specific parts of beef carcasses and surfaces of the processing environment at the slaughterhouse of Ha′il, Saudi Arabia between September and November 2020. MALDI BioTyper and phylotype-based identification methods accurately identified and classified the samples as belonging to the genus belonging to the *Escherichia coli* domain of bacteria (NCBI txid: 562). Results: Expected changes were seen in the mass peak spectrum defining nine closely related isolates and four unrelated *E. coli* isolates. Serological typing of *E. coli* revealed enterotoxigenic *E. coli* O166 (19.10%); enteropathogenic *E. coli* O146 (16.36%) and O44 (18.18%); enterohemorrhagic *E. coli* O111 (31.18%) and O26 (14.54%). Forty-five percent of examined *E. coli* were resistant to seven antimicrobials; 75% of 20 selected isolates were resistant to three or more antimicrobials. *phoA* and *bla_TEM_* genes were detected in all selected *E. coli* isolates. Conclusion: This study confirmed the efficiency and validity of Matrix-assisted laser desorption/ionization time of flight mass-spectrometry in screening for multi-drug resistant *E. coli* isolated from slaughterhouse derived beef carcasses in Ha’il, Saudi Arabia. We contributed by revealing the distinction between related and non-related strains of *E.*
*coli* in livestock. The findings in this study can inform improved policy development decision making and resource allocation related to livestock processing regarding antimicrobial use in food animals and rapid screening for effective multiple antibiotic resistance *E. coli* and virulent non-O157 STEC control in the slaughterhouses.

## 1. Introduction

Shiga toxin-producing *Escherichia coli* (STEC) remain a significant food safety issue in raw meat and meat products. Despite the introduction of mandatory testing for STEC in beef trim and use of multi-level intervention strategies, there are still sporadic outbreaks of foodborne illnesses with product recalls linked to STEC contamination. There is a growing public health concern of STEC contamination posing as a threat to the meat industry globally. For instance, STEC contamination of meat and meat products in the Germany 2011 outbreak resulted in over 4000 cases of illnesses [1].

STEC have been implicated in a few human gastrointestinal ailments such as non-bloody and bloody diarrhea complicated by hemolytic uremic syndrome (HUS). STEC serotype O157:H7 is the main cause for HUS [2]. However, non-O157 STEC have been reported recently as reasons for bloody diarrhea and HUS in Canada, United (US), and Europe [2]. There are six serogroups of STEC commonly linked to clinical infections. The top six serogroups in the US including O26, O45, O103, O111, O121, and O145 have been recently classified as contaminants in beef [3]. Non-O157 STEC are endemic in herds of cattle, sheep, and goats [4,5]. Ruminants are asymptomatic carriers of STEC in their intestinal floral. Cattles specifically are thought to be the main carriers of non-O157 STEC [5]. Surveys revealed that 0.4–74% of manure screened from dairy or beef farms tested positive for non-O157 STEC [5].

### Literature Review 

Prior studies have revealed that the processes of unhygienic carcass handling have led to the contamination of hides, unpasteurized milk, raw meat, and some meat products by STEC. In addition, surrounding environments have been polluted through poor disposal of STEC contaminated intestinal content and feces of animals [2]. The continued occurrence of *Escherichia coli* (*E. coli*) O157:H7 in beef can be attributed to its high prevalence at the primary phase of beef production, and poor implementation of current carcasses decontamination interventions required during beef production. Most public health and clinical laboratories test only for O157 STEC in humans. There is a dearth of data on the proportion of non-O157 STEC [2].

*Escherichia coli* (*E. coli*) O157 laboratory screening is based on its inability to ferment sorbitol. Detection of non-O157 STEC is problematic due to the lack of reliable selective agars and its propensity to ferment sorbitol, making its identification difficult compared to O157 STEC which does not ferment sorbitol [6]. Unlike *E. coli* O157, non-O157 STEC express glucuronidase and ferment sorbitol, hence appear the same as generic *E. coli* on CT-SMAC [7]. Nevertheless, non-O157 STEC share the same antibiotic resistance as O157 STEC so some selective enrichment is possible. Identification of the top six serotypes is achieved through immunomagnetic separation (IMS) and PCR-based techniques [8,9,10]. Molecular compared to culture-based methods are more advantageous in terms of specificity, rapidity, and high throughput [11]. Surveys performed to date would suggest that non-O157 STEC share the same habitats and dissemination routes as O157 STEC [5,12]. The lack of reliable diagnostic techniques for detecting non-O157 STEC, particularly for isolation, has hampered progress in estimating the actual prevalence of these bacteria. Quick and explicit identification of foodborne microbes are fundamental parts of food safety and resulting execution of successful control measures. While conventional molecular typing techniques are time-consuming in generating typing data [13], Matrix-assisted laser desorption/ionization time of flight mass-spectrometry (MALDI-TOF MS) is more efficient in determining precise protein mass-spectra profiles, thus useful for assessing strain relatedness [13]. In this study, we aimed to assess the validity of MALDI-TOF MS based typing for rapid screening of multiple antibiotic resistance *E. coli* and virulent non-O157 shiga toxin-producing *E. coli* isolated from beef carcasses in a slaughterhouse setting.

## 2. Materials and Methods

### 2.1. Sampling and Sample Collection

A total of 180 convenience samples were collected from two different sources-specific parts of beef carcasses and surfaces of the processing environment at the slaughterhouse of Ha′il, Saudi Arabia between September and November 2020. These sources were based on previous research on the dissemination of enteric contamination within beef processing operations [14] This study identified the de-hiding step as a critical point along with the evisceration process as sources of *E. coli* contamination. Carcass samples were taken after de-hiding and post-evisceration. The samples were collected at three different carcass locations—shoulder, brisket, and thigh using a sterile sponge. Environmental samples were obtained from various surfaces including (knives and workers′ hands and holding areas of the processing line (slaughterhouse floor, wall, and effluent). After sampling, sponges were placed into sterile stomacher bags and stored at 4 °C until processing [14].

### 2.2. Isolation and Identification of Bacterial Strains

Analyses of samples were carried out according to ISO/CEN 13136:2012 (ISO/TS 13136, 2012). Each sponge was placed into a stomacher bag containing 500 mL of modified trypticase soy broth containing 8 mg/L novobiocin. Each sponge was blended for two minutes and incubated for 20 h at 41.5 °C. One ml of each sample was cultured on MacConkey agar medium and Levine-eosin methylene blue agar (Himedia, Mumbai, India) and followed by incubation at 37 °C. Sorbitol fermentation was tested on sorbitol MacConkey agar and sorbitol phenol red agar media (Himedia, Mumbai, India) (overnight incubation at 37 °C).

Identification of bacterial strains was performed by using conventional methods. Further identification was confirmed by MALDI-TOF MS and Microscan, according to the manufacturer’s guidelines [15]. A single colony of a subculture was directly deposited in duplicate on a MALDI-TOF-MS plate (Bruker Daltonik GmbH, Karlsruhe, Germany) and the results were noted. Serological identification was completed by utilizing demonstrative polyvalent and monovalent *E. coli* antisera based on the instructions of the manufacturer (BIO-RAD, Marnes-la-Coquette, France) [2]. To validate bacterial species using the library database, revealed profiles were first analyzed. To detect single peaks of each isolate, Flex Analysis software (Bruker Daltonics, Karlsruhe, Germany) was used. Furthermore, the peak changes were found in an overlay of all isolates to assess non-related clusters. Thereafter, the profiles were smoothed and using the BioTyper 3 program, baseline peak changes were deducted. The MALDI BioTyper 3 software (Bruker Daltonics, Karlsruhe, Germany) was used to convert the obtained spectra into a virtual gel (pseudo-gel-like) format. This virtual gel view represented all of the peaks in a spectral file and was used to compare the spectra of the *E. coli* isolates that were tested. Clusters with similar protein expression were identified by the principal component analysis (PCA) [13].

### 2.3. Antimicrobial Resistance

Anti-microbial resistance (AMR) was determined against 14 antimicrobials, for 20 selective isolates from carcass shoulder, brisket, and thigh and from abattoir effluent (5 from each represent the serotypes O166, O146, O44, O111, and O26), using the disc diffusion method using *E. coli*-ATCC 25,922 as a reference strain according to Clinical and Laboratory Standards Institute (CLSI) guidelines [16]. Antimicrobial discs were used: penicillin (P) (10 UI), erythromycin (E) (15 μg), oxytetracycline (T) (30 μg), nalidixic acid (NA) (30 μg), ampicillin (AM) (10 μg), sulfamethoxazole (SXT) (23.75 μg), cephalothin (CN) (30 μg), enrofloxacin (EX) (5 μg), oxacillin (OX) (1 μg), neomycin (N) (30 μg), chloramphenicol (C) (30 μg), kanamycin (K) (30 μg), ciprofloxacin (CP) (5 μg), gentamicin (GEN) (10 μg). Inoculum of each strain was streaked on Mueller–Hinton agar (Himedia, Mumbai, India), and the appropriate drug-impregnated discs were placed on the agar surface.

Multiple antibiotic resistance (MAR) index was investigated. MAR index is an instrument to examine wellbeing and health hazards. This index is useful for checking the spread of bacterial resistance in a given population where there is resistance to more than three antibiotics [17]. The MAR index is determined by the quantity of antibiotics to which test serotypes we found to show resistance divided by all antibiotics used for sensitivity assessment. Estimated MAR index of more than 0.2 indicates an environment susceptible to high risk of contamination and antibiotics use.

### 2.4. PCR Assay

Polymerase chain reaction (PCR) monitoring of *phoA* virulence-determinant gene and the antibiotic-resistance genes *bla_TEM_* was carried out as described by some authors [18,19]. In relation to the QIAamp DNA Mini Kit (Qiagen, GmbH, Germany/Catalog No.51304) manufacturer guidelines, genomic DNA of the examining strains were extracted. The primer pairs used have been mentioned in previous studies [19,20].

### 2.5. Statistical Analysis

SPSS program, version 26, was used to perform the Chi-square test (significance level: *p* < 0.05).

## 3. Results and Discussion

Contamination of the environment in slaughterhouses with *E. coli* may be due to bowel rupture during evisceration, indirect contamination with tainted water, handling and packaging of finished products [21]. *E. coli* were detected in 80%, 65%, 75%, 100%, 55%, 45%, 30%, and 100% of examined beef shoulder, beef thigh, beef brisket, floor, wall, knives, worker hands, and effluent, respectively (Table 1). The isolation and identification of serotypes were carried out by conventional methods and compared by rapid detection using MALDI-TOF MS.

All spectra were magnified and mass peak profiles tested to detect typical *E. coli* isolates peaks. *E. coli* isolates were reliably identified by MALDI BioTyper and achieved scores of >2.3 (highly probable identification of organisms) [22]. MALDI BioTyper and phylotype-based identification methods accurately identified and classified the sponge samples as belonging to the genus belonging to the *Escherichia coli* domain of bacteria (NCBI txid:562).

The validity of peak patterns as taxonomic markers is set up utilizing MALDI-TOF MS as a bacterial fingerprinting method. Findings showed that these criteria are met in the samples we tested in this analysis. MS profiles are reliable when a group of isolates is studied simultaneously under the same experimental and instrumental conditions, ensuring high repeatability and reproducibility across studies. Mass spectra, on the other hand, seem to have an inherent variability. All the tested isolates had mass peak profiles between 2000 and 12,000 m/z ratio. To visually distinguish spectra, the five discriminatory regions were extended. The ability of MALDI-TOF MS to discriminate may be used to screen *E. coli* strains isolated from beef carcasses and slaughterhouses. This indicates a strong association with other methods of identification and stronger diagnostic tests.


Expected changes were seen in the mass peak spectrum (Figure 1). Mass peaks were dramatically shifted to allow separation of non-related clusters (Figure 1). Principal component analysis (PCA) showed that the clustering was clarified by PC1, PC2, and PC3. Figure 2 summarizes the findings of a two-dimensional cluster analysis. A high resolution to discriminate against non-related clusters was revealed by plotting PC1 and PC2.

The results shown in Figure 3 and Figure 4 explain the relationship and source of contamination between the isolated serotypes from beef shoulder (five serotypes: O166, O146, O44, O111, and O26) and isolated serotypes from knives (four serotypes: O166, O44, O111, and O26) and workers′ hands (four serotypes: O166, O146, O44, and O111), which were all tested at the same time of sampling. Clusters were precisely delineated by a PCA-based dendrogram and virtual gel view (Figure 3 and Figure 4), defining nine closely related isolates and four unrelated *E. coli* isolates. Thus, the results obtained from virtual gel analysis and PCA confirmed our findings. In less than one day, all results from MALDI-TOF MS, including the thorough analysis of peak frame changes, were obtained. Isolates representing related *E. coli* strains were unclear from one another but were delineated from non-related strains. In addition, Figure 3 and Figure 4 revealed highly distinguishable diverse peaks within non-related *E. coli* isolates. To the best of our knowledge, this is the first study to highlight the distinction between related and non-related strains of *E. coli* in livestock. However, further studies are needed to substantiate this finding.

The study was effective in identifying the antimicrobial resistance and the prevalence of virulent non-O157 STEC serotypes in meat and the slaughterhouse settings. The MALDI-TOF MS technique, on the other hand, was used to identify certain species in a short amount of time, in tandem with a study of the relationship between the distribution of those species in various sources, as revealed by the results of principal component analysis (PCA) and virtual gel analysis for related and non-related clusters. The ability of MALDI-TOF MS to discriminate *E. coli* strains isolated from beef carcasses and slaughterhouses could be used to screen them. This points to a close link between other methods of identification and more powerful diagnostic tests. These findings suggest that these isolates have the same origin, necessitating vigilance and the development of policies and strategies regarding rapid screening for effective multiple antibiotic resistance
*
E. coli
*
and virulent non-O157 STEC control in slaughterhouses and the application of approved hygienic procedures.


Occurrences of *E. coli* in slaughterhouses samples have been recorded worldwide [23,24]. *E. coli* have a biphasic nature and can live effectively either in the environment or in the host, where animal excreta and human sewage are released. Pathogenic *E. coli* of zoonotic importance, (enterohemorrhagic, enteropathogenic, enterotoxigenic) can also be accessed from slaughterhouse discharges in the environment. Slaughterhouse livestock effluents—blood and fecal matter are excellent sources for bacterial growth and multiplication. Waste material removed from cleaning slaughterhouses with water and may contaminate the surrounding environment if effluents are poorly handled. Serological typing of *E. coli* revealed enterotoxigenic *E. coli* O166 (19.10%); enteropathogenic *E. coli* O146 (16.36%) and O44 (18.18%); enterohemorrhagic *E. coli* O111 (31.18%) and O26 (14.54%) (Table 2). These results are similar to the serotypes detected in slaughterhouse samples from Namibia [23].

This research tested the antimicrobial resistance of 20 *E. coli* isolates, selected from carcass shoulder, brisket, and thigh and from abattoir effluent (5 from each represent the serotypes O166, O146, O44, O111, and O26), against 14 widely used antimicrobials to detect trends of resistance associated with them. In Table 3, all tested isolates (*20 selected isolates)* were 100% resistant to penicillin, followed by erythromycin (80%), oxytetracycline (75%), nalidixic acid (65%), ampicillin (60%), sulfame-thoxazol (55%), cephalotin (45%), enroflo-xacin (40%), oxacillin (35%), neomycin (30%), chloramphenicol (20%), and kanamycin (15%).The lowest resistance was against gentamicin (10%) and ciprofloxacin (5%). The antimicrobial susceptibility patterns of *E. coli* found in our samples were similar to previously published research by [25]. In this study, we detected *E. coli* resistance to ceftriaxone (4.44%), chloramphenicol (4.44%), ciprofloxacin (2.22%), gentamicin (2.22%), suphamethoxazole/trimethoprim (17.78%), and tetracycline (28.89%). This result is similar to the study conducted in Ghana [26].

The MAR index was of 0.533 (0.071 to 1.000). Forty-five percent of examined *E. coli* were resistant to seven or more antimicrobials (Table 4). MAR patterns showed that 75% of the isolates were resistant to three or more antimicrobials. This is similar to findings in prior research [26]. The proportion of the isolates with MAR index, more than 0.2 was 75%, and less than or equal to 0.2 was 25%. MAR index value higher than 0.2 indicates high-risk sources of contamination, where several antimicrobials may often be used for the control of diseases [27]. The higher resistance of *E. coli* isolates could be attributed to the misuse of antibiotics for therapeutic or wide use as growth promoters among livestock.

As the spread of multidrug-resistant (MDR) bacteria has been repeatedly warned of, resistance to several antimicrobials poses a threat to human health. The findings of multiple antibiotic resistance (MAR) index and antimicrobial resistance profile of the isolated *E. coli* strains confirm the need for development of policies and strategies regarding antimicrobial use in food animals.

The presence of virulence, and antibiotic-resistance genes of *E. coli* isolated from beef carcass and abattoir environment samples was investigated using a PCR assay of the virulence-determinant gene (*phoA*) and the antibiotic-resistance genes *bla_TEM_*. For the PCR assay, serotypes isolated from beef shoulder and abattoir effluent (O166, O146, O44, O111, and O26) were chosen. *E. coli phoA* gene has been extensively studied and used as a marker to recognize secreted proteins, because it is an exported enzyme that is only activated in the bacterial periplasmic space after its translocation. Presence of *E. coli phoA* gene increased the intimin protein expression and increased *eaeA* mRNA production. Hence, *phoA* gene is considered essential for virulence from Enteropathogenic *E. coli* [28]. The alkaline phosphatase gene (*phoA*) is used in all *E. coli* strains and has previously been used in PCR-based methods for strain detection with high specificity. As a result, the *phoA* gene was chosen as a target for identification of all *E. coli* strains in this analysis (both related and non-related strains).

The PCR results of some representative isolates in Figure 5 revealed that *phoA* gene was detected at 720 bp in all examined *E. coli* isolates, indicating elevated pathogenicity to host epithelial cells. The *phoA* gene was found in *E. coli* from bovine feces [20].

Antimicrobial resistance has become a major public health problem worldwide. Given the prevalence of MDR bacteria, therapeutic options for many infectious diseases are currently limited. β-lactam antibiotics are commonly used in humans and in veterinary medicine to treat bacterial infections [20]. A large number of β-lactamase producers have emerged in Gram-negative bacteria since the introduction of beta-lactam antibiotics, especially in Enterobacteriaceae, such as *E. coli*. The *bla_TEM_* is common in non-O157 STECs in animals, and consequently, the high rates of ampicillin-resistant *bla_TEM_* -positive isolates were to be expected. The *bla_TEM_* was identified in non-O157 STEC isolated from cattle feces and soil samples, but none of the other *bla_OXA_, bla_SHV_*, and *blaC_TX-M_* variants could be detected. Poultry-associated STEC is more likely to carry the *bla_SHV_* variant, and *bla_OXA_*, *bla_SHV_*, and *bla_CTX-M_* have been previously reported in STEC O157:H7 Serotypes. As a result, the *bla_TEM_* gene was chosen as a target for identification of non-O157 STEC strains isolated from beef carcasses and slaughterhouse settings.

The PCR results of some representative isolates revealed that the *bla_TEM_* gene was detected at 516 bp in all examined *E. coli* isolates (Figure 6). Previously detected genes in *E. coli* were from, lamb meat (97.1%), meat products (69.2%,), and retail chicken 28.1% [29,30,31,32,33].

The data available in Saudi Arabia revealed the presence of O157:H7 in camel, sheep, goat, and cattle feces and hides in Riyadh [34]. Raw meat and milk samples were found to be contaminated with O157:H7 STEC and non- O157 STEC (O111, O103, and O22) from different locations in Riyadh [2]. The prevalence in large variations may be related to screening methods employed in studies using molecular-based methods to detect virulence genes resulting in higher values when compared to utilizing isolation-based methods [35]. While cattle harbor a diverse range of non-O157 STEC, given that the majority lack the essential virulence factors, only 12–17% of these serotypes have been isolated from serious cases of illness in humans [36]. Similar to O157 STEC, shedding of non-O157 STEC is seasonal with high shedding occurring in spring through fall compared to the winter months [37]. The majority of cattle acquire non-O157 STEC from contact with infected animals or indirectly via consumption of fecal-contaminated feed and water [38]. The prevalence of both groups of STEC was reduced by post-evisceration interventions that included steam vacuuming, trimming, steam pasteurization, and organic acid wash. The carriage of virulent non-O157 STEC can be reduced by interventions applied during the slaughter process. The difference in prevalence of non-O157 STEC between studies may partly be due to application of different study design methods. The pervasiveness of non-O157 STEC in beef carcasses has the potential to be equivalent or more prominent than that of O157 STEC. Prevalence rates of all STEC in beef samples were mostly greater than 20% (15–40%) [39].

This study is not without limitation. It is a cross-sectional study thus does not show temporality and causation. The non-probability sampling method employed (convenience sampling) restricts generalizability of our findings to the whole of Saudi Arabia. Advanced hygienic procedures during slaughtering may be critical in limiting the spread of multiple antibiotic resistance *E. coli* as well as lowering contamination levels in the environment.

## 4. Implications for Practice and Theory

This study, similar to previous studies, highlights the benefits of using molecular methods for detecting virulent *E. coli* as opposed to isolation-based method for detecting *E. coli* in livestock. Furthermore, distinguishing between related and non-related strains of *E. coli* can further improve the taxonomy of pathogenic *E. coli*. in livestock.

Non-0157 STEC are not as virulent as 0157 STEC. Thus, proper identification and classification of pathogenic and non-pathogenic *E. coli* strains is paramount to avoid wastage of edible livestock.

Further research into developing substitute growth promoters other than antimicrobial is needed in order to prevent MDR occurrence in livestock and subsequent zoonotic infections in humans.

Policy development, decision making, and resource allocation regarding screening for *E. coli* contamination in livestock and provision of health care services for livestock-induced *E. coli* related illnesses in humans can be improved by using data driven knowledge of seasonal shedding of *E. coli* in livestock.

Promotion of necessary and sufficient post-evisceration interventions and other good slaughter practices are required to stem livestock-induced *E. coli* in humans.

## 5. Conclusions

The study was effective in identifying the prevalence of multiple antibiotic resistance *E. coli* and virulent non-O157 STEC serotypes in meat and slaughterhouse settings. Anti-microbial resistance (AMR) was determined against 14 antimicrobials and multiple antibiotic resistance (MAR) index was investigated. The MALDI-TOF MS technique was used to discriminate *E. coli* strains isolated from beef carcasses and slaughterhouses, with a study of the relationship between the distribution of those species in various sources, as revealed by the results of principal component analysis (PCA) and virtual gel analysis. *phoA* and *bla_TEM_* genes were detected in all selected *E. coli* isolated from beef shoulder and abattoir effluent. These findings suggest that these isolates have the same origin, necessitating vigilance and the development of policies and strategies regarding antimicrobial use in food animals and rapid screening for effective multiple antibiotic resistance *E. coli* and virulent non-O157 STEC control in slaughterhouses and the application of approved hygienic procedures.

## Figures and Tables

**Figure 1 foods-10-00820-f001:**
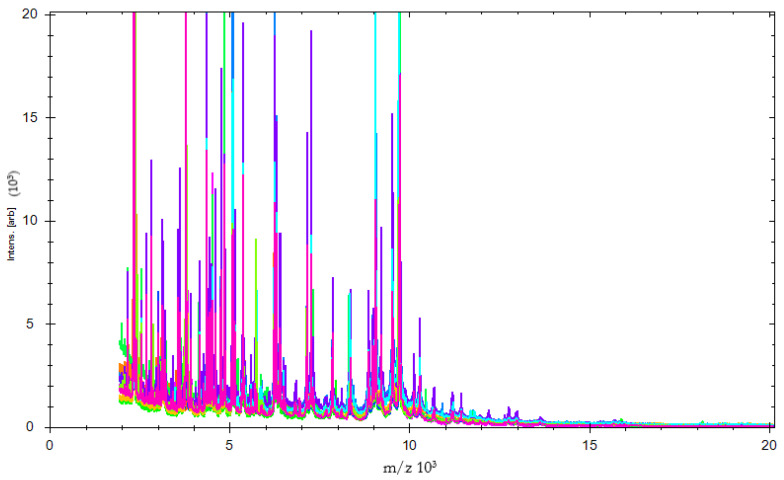
Characteristic spectra of MALDI-TOF MS. The variations in the mass spectrum of *E. coli* isolates are displayed by color.

**Figure 2 foods-10-00820-f002:**
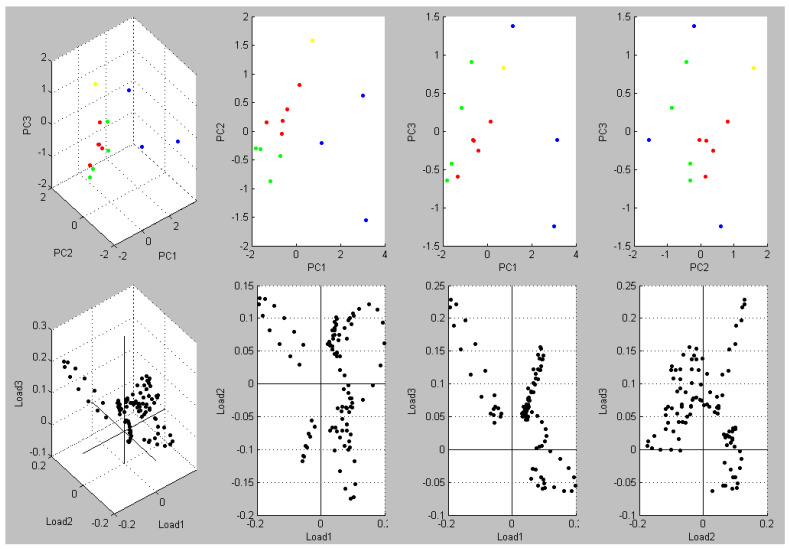
Three-dimensional principal component analysis of MALDI-TOF MS mass spectra of the tested *E. coli* isolates. The greatest potential for distinction was shown by PC1 and PC2 and a cluster of isolates was shown, whereas the non-related isolates are less similar.

**Figure 3 foods-10-00820-f003:**
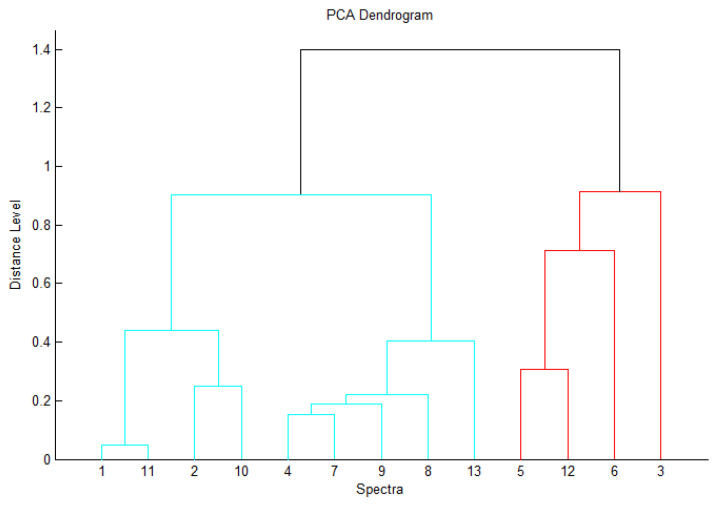
PCA-based dendrogram mass spectra of the tested *E. coli* isolates generated by MALDI-TOF MS. Non-related isolates are illustrated by red.

**Figure 4 foods-10-00820-f004:**
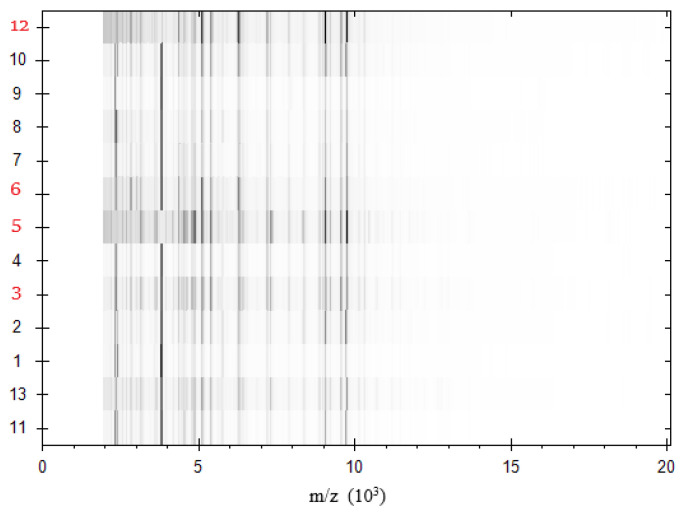
Virtual gel analysis using the MALDI BioTyper 3′s special tool reveals changes in band pattern. The m/z values are represented on the x axis, and the obtained mass spectra of the tested *E. coli* isolates are represented on the y axis. The protein expression profile peaks produced by MS are seen for isolates. Non-related isolates are illustrated by red labels.

**Figure 5 foods-10-00820-f005:**
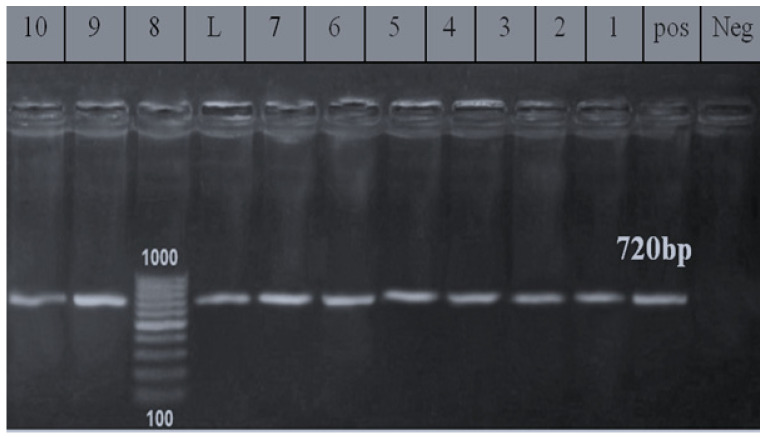
Agarose gel electrophoresis of *phoA* gene (720 bp). L: 100 bp ladder as molecular size DNA marker. Pos: Control positive for *phoA* gene. Neg: Control negative. Lanes from 1 to 10: Positive *E. coli* strains (O166, O146, O44, O111, and O26) isolated from beef shoulder and abattoir effluent *for phoA* gene.

**Figure 6 foods-10-00820-f006:**
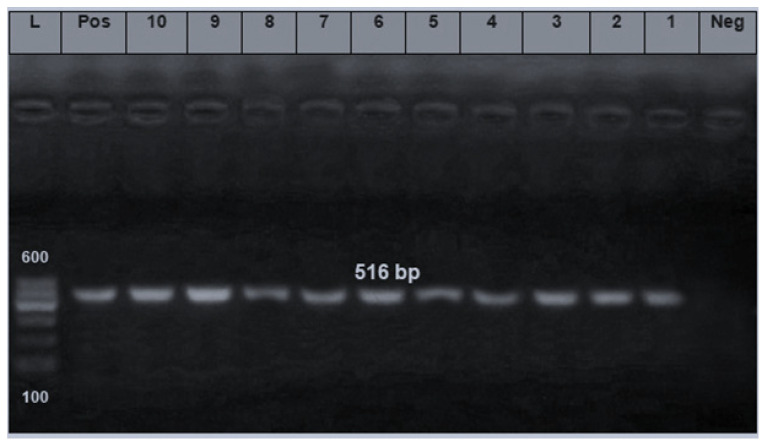
Agarose gel electrophoresis of *bla_TEM_* gene (516 bp). L: 100 bp ladder as molecular size DNA marker. Pos: Control positive for *bla_TEM_* gene. Neg: Control negative. Lanes from 1 to 10: Positive *E. coli* strains (O166, O146, O44, O111, and O26) isolated from beef shoulder and abattoir effluent for *bla_TEM_* gene.

**Table 1 foods-10-00820-t001:** Prevalence of *E. coli* in different kinds of samples taken from beef carcasses and slaughterhouse environments (*n* = 20).

Samples	Prevalence
Abattoir effluent	20 (100%)
Abattoir floor	20 (100%)
Abattoir wall	11 (55%)
Beef brisket	15 (75%)
Beef shoulder	16 (80%)
Beef thigh	13 (65%)
Knives	9 (45%)
Water	Not detected
Worker hands	6 (30%)

**Table 2 foods-10-00820-t002:** Serotyping of *E. coli* strains isolated from beef carcasses and slaughterhouse environments samples.

	O166	O146	O44	O111	O26
Abattoir effluent	3	2	4	8	3
Abattoir floor	3	4	5	6	2
Abattoir wall	2	3	2	2	2
Beef brisket	4	2	3	3	3
Beef shoulder	2	4	1	7	2
Beef thigh	3	1	2	4	3
Knives	3	-	2	3	1
Worker hands	1	2	1	2	-
Total	21 (19.10%)	18 (16.36%)	20 (18.18%)	35 (31.81%)	16 (14.54%)
Types	ETEC	EPEC	EPEC	EHEC	EHEC

EHEC: *Enterohemorrhagic E. coli*. EPEC: *Enteropathogenic E. coli*. ETEC: *Enterotoxigenic E. coli*. *E. coli*: *Escherichia coli*.

**Table 3 foods-10-00820-t003:** Antimicrobial resistance pattern of the isolated *E. coli* strains from carcass shoulder, carcass brisket, carcass thigh, and abattoir effluent (*n* = 20).

Antimicrobial Agent	Sensitive	Intermediate	Resistant	Serotype	*n*	Pathotype
Penicillin (P)	-	-	20 (100%)	O166 O146 O44 O111 O26	5 5 5 5 5	ETEC EPEC EPEC EHEC EHEC
Erythromycin (E)	-	2 (20%)	18 (80%)			
Oxytetracycline (T)	3 (15%)	2 (10%)	15 (75%)			
Nalidixic acid (NA)	4 (20%)	3 (15%)	13 (65%)			
Ampicillin (AM)	-	8 (40%)	12 (60%)			
Sulfamethoxazol (SXT)	6 (30%)	3 (15%)	11 (55%)			
Cephalotin (CN)	9 (45%)	2 (10%)	9 (45%)			
Enrofloxacin (EN)	10 (50%)	2 (10%)	8 (40%)			
Oxacillin (OX)	12 (60%)	1 (5%)	7 (35%)			
Neomycin (N)	14 (70%)	-	6 (30%)			
Chloramphenicol (C)	16 (80%)	-	4 (20%)			
Kanamycin (K)	15(75%)	2 (10%)	3 (15%)			
Ciprofloxacin (CP)	16 (80%)	2 (10%)	2 (10%)			
Gentamicin (G)	19 (95%)	-	1 (5%)			
*p* value	*p* < 0.0001	*p* < 0.0001	*p* < 0.0001			

ETEC, *enterotoxigenic E. coli*; EPEC, *enteropathogenic E. coli*; EHEC, *enterohemorrhagic E. coli*.

**Table 4 foods-10-00820-t004:** Multiple antibiotic resistance (MAR) index and antimicrobial resistance profile of the isolated *E. coli* strains from carcass shoulder, carcass brisket, carcass thigh, and abattoir effluent (*n* = 20).

Resistance Pattern	Resistance Profile	Number of Isolates	Number of Antibiotics	MAR
i.	P, E, T, NA, AM, SXT, CN, EN, OX, N, C, K, CP, G	1	14	1
ii.	P, E, T, NA, AM, SXT, CN, EN, OX, N, C, K, CP	1	13	0.92
iii.	P, E, T, NA, AM, SXT, CN, EN, OX, N, C, K	1	12	0.85
iv.	P, E, T, NA, AM, SXT, CN, EN, OX, N, C	1	11	0.78
v.	P, E, T, NA, AM, SXT, CN, EN, OX, N	2	10	0.714
vi.	P, E, T, NA, AM, SXT, CN, EN, OX	1	9	0.642
vii.	P, E, T, NA, AM, SXT, CN, EN	1	8	0.571
viii.	P, E, T, NA, AM, SXT, CN	1	7	0.5
ix.	P, E, T, NA, AM, SXT	2	6	0.428
x.	P, E, T, NA, AM	1	5	0.357
xi.	P, E, T, NA	1	4	0.285
xii.	P, E, T	2	3	0.21
xiii.	P, E	3	2	0.142
xiv.	P	2	1	0.071
Average				0.533

## Data Availability

The datasets generated and/or analyzed during the current study are available from the corresponding author on reasonable request.

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
