# Peer review of "MALDI-TOF MS Based Typing for Rapid Screening of Multiple Antibiotic Resistance E. coli and Virulent Non-O157 Shiga Toxin-Producing E. coli Isolated from the Slaughterhouse Settings and Beef Carcasses"

_foods, 2021, doi:10.3390/foods10040820_

Round 1

Reviewer 1 Report

  1. The author is requested to clearly explain the current policy-making and decision-making issues related to the allocation of resources related to livestock processing, and what the Information and the finding provided in this study is and should specifically suggest improvements and future directions. Base upon the Line 40 emphasize that the research findings can be a reference for improved processing-related resource allocation and future policy making.
  2. Line 103-104, The samples were collected at three different carcass locations-shoulder, breast and leg using a sterile sponge. However, Table 1 and Table 2 are information on only beef shoulder.
  3. In the paragraph 2.3. Antimicrobial resistance, this study requires a reference strain and QC range, and the author should explain why they are missing. The author can refer to this document as following, CLSI, Clinical and Laboratory Standards Institute: Performance standards for antimicrobial susceptibility testing: 27th ed 424 informational supplement. CLSI Doc. M100-S20 (2017). Also some USFDA antimicrobial susceptibility test guidance.
  4. The 14 testing antimicrobial agents were used but lack the information about the concentrations or the amount the agents.
  5. In the section of 2.4. Molecular typing, only the phoA virulence-determinant gene and the antibiotic-resistance genes blaTEM were selected in this research. Base on the section title “Molecular typing” there were lack the typing step in the study, only 2 kinds PCR test and with results without any typing and analysis technology. Further, the virulence-determinant genes were used as test tools not only phoA but also tsh, hly, eaeA, sta, lt… etc. And the blaTEM was used as antibiotic-resistance genes test, there are also blaCTX, blaKPC… etc. could be used. Suggestion for the author, the selection of these antibiotic-resistance genes would be based upon the results of the antimicrobial susceptibility testing, moreover, the selection of the current virulence genes based upon their signifcant role in the pathogenesis of the disease, that all on your test bacterial strains characteristic in this research. Basically, the phoA gene is considered to be as a marker to recognize secreted proteins, also critical to the virulence of enteropathogenic E. coli, but most of the E. coli in this study belong to ETEC and EHEC. Can't just only choose phoA gene used as marker. In a similar situation, the blaTEM gene that one of β-lactamase producers. It is not enough to only use the blaTEM for testing.
  6. Some data presentation methods will mislead readers. In this research tested the antimicrobial resistance of 20 E. coli isolate the results in stable 3, all tested isolates were 100% resistant to Penicillin. Readers may misunderstand that all E. coli isolates have resistance to Penicillin.
  7. The research results in Table 1 and Table 2 show that the author has at least 110 different E. coli strains in this study. However, there are only 20 E. coli strains for antimicrobial susceptibility. The author needs to explain the relationship and differences between these 20 strains and the 110 strains. Why did the 110 strains not be tested for Antimicrobial susceptibility? How the 20 strains were selected and decided, and the representativeness of the entire study should also be explained clearly.
  8. In the experimental result analysis table in Table 3, these strains are independent and different, and the properties and conditions of the 14 antibiotics tested are also different. Therefore, the minimum requirement statistical analysis methods should be used to show the correlation or error of the result data.
  9. At the same time, it is suggested that the author add two columns to Table 3, namely serotype and disease type, so that it can show the relationship between serotype and pathogenic E. coli to the antibiotic resistance selected in this study.
  10. In the results of antibiotic resistance pattern and MAR index, MAR and final average values have been generated, but the results of individual antibiotic response are different. Please explain how to benchmark and statistical methods of individual antibiotic test results, so that readers will be published in the future easy to understand.
  11. The variations in the mass spectrum of E. coli isolates are showed with the representative example of typing dependent on MALDI-TOF MS. However, there are that the author did not explain and analyze the typing dependent data results shown.
  12. The author uses principal component analysis to strengthen the presentation of experimental results and reduce the influence of interference factors on the results. PC1 and PC2 showed the highest discrimination potential and showed a cluster of isolates, while the unrelated isolates were not very similar. Especially in Figure 2, 3 and Line 181-189. The related and unrelated isolates that the definition and difference of the related and unrelated isolates should be explained here, so that readers can understand their differences and importance. But it is a pity that the author did not explain, analyze and interpret the results shown. The author must explain, analyze and discuss these important experimental results, so that readers can understand the spirit of these research experiments and the value of research results.
  13. For the isolate, the protein expression peak produced by MS can be seen. Irrelevant isolates are marked in red in Fig. 4. However, in the chapter on materials and methods, Virtual gel is missing. Please add the source of experimental materials, well/ Lane concentration, electrophoresis method, and analysis parameters of electrophoresis graphs, in order to complete the readers to understand its differences and comparison benchmarks.
  14. The source of the 10 strains of coliform bacteria used in Figure 5 and Figure 6 has not been explained, and the significance of the two experimental results is not clear. The author is requested to provide a clear explanation and description. Line 246-277 has explained that the detected phoA and blaTEM gene in E. coli have different detection rates from different meat sources. At this moment, only 10 strains are shown in the 2 figures, and all the detected strains are shown. The author has fully and detailed explanation and analysis are necessary so as not to misunderstand readers.
  15. This research is very commendable for analyzing and discussing the characteristics of meat, work equipment and environmental E. coli, but it failed to study and confirm the relationship between these strains collected from different sources. For example, an E. coli was detected separately in the intestines, meat, utensils, personnel, and environment/waste water during the manufacturing process, and it was confirmed that it was the transmission of the same strain. It is very pity that Line 292-295 failed to specifically address the policy formulation, decision-making and resource allocation of health care services. For the prevention of MDR occurrence in livestock and avoiding the spread of slaughter or food to humans, it failed to point out the priorities and control methods in Line 289-291.

Author Response

Response to Reviewer 1 Comments

****

Point 1: The author is requested to clearly explain the current policy-making and decision-making issues related to the allocation of resources related to livestock processing, and what the Information and the finding provided in this study is and should specifically suggest improvements and future directions. Base upon the Line 40 emphasize that the research findings can be a reference for improved processing-related resource allocation and future policy making.

Response 1:

The findings in this study can inform improved policy development decision making and resource allocation related to livestock processing regarding antimicrobial use in food animals and rapid Screening for effective MDR-E. coli and virulent non-O157 STEC control in the Slaughterhouses. See line 42

Point 2: Line 103-104, The samples were collected at three different carcass locations-shoulder, breast and leg using a sterile sponge. However, Table 1 and Table 2 are information on only beef shoulder.

Response 2: Done

The samples were collected at three different carcass locations- shoulder, brisket and thigh (see line 106).  

Point 3:  In the paragraph 2.3. Antimicrobial resistance, this study requires a reference strain and QC range, and the author should explain why they are missing. The author can refer to this document as following, CLSI, Clinical and Laboratory Standards Institute: Performance standards for antimicrobial susceptibility testing: 27th ed 424 informational supplement. CLSI Doc. M100-S20 (2017). Also, some USFDA antimicrobial susceptibility test guidance.

Response 3: Done

Anti-microbial resistance (AMR) was determined against 14 antimicrobials for each isolate using the disc diffusion method using E. coli-ATCC 25922 as a reference strain according to CLSI guidelines [16].  (See lines 135, 136 & 137).

 The document, CLSI, was cited before with reference No. 16

Point 4: The 14 testing antimicrobial agents were used but lack the information about the concentrations or the amount the agents.

Response 4:  Done (see lines 138 to 142)

Antimicrobial discs were used: Penicillin (P) (10 UI), Erythromycin (E) (15 μg), Oxytet-racycline (O) (30 μg), Nalidixic acid (NA) (30 μg), Ampicillin (AM) (10 μg), Sulfa-methoxazole (SXT) (23.75 μg), Cephalothin (CEP) (30 μg), Enrofloxacin (EX) (5 μg), Oxacillin (OX) (1 μg), Neomycin (N) (30 μg), Chloramphenicol (C) (30 μg), Kanamycin (K) (30 μg), Ciprofloxacin (CIP ) (5 μg), Gentamicin (GEN) (10 μg).  See line 135 to 139)

Point 5:   In the section of 2.4. Molecular typing, only the phoA virulence-determinant gene and the antibiotic-resistance genes blaTEM were selected in this research. Base on the section title “Molecular typing” there were lack the typing step in the study, only 2 kinds PCR test and with results without any typing and analysis technology. Further, the virulence-determinant genes were used as test tools not only phoA but also tsh, hly, eaeA, sta, lt… etc. And the blaTEM was used as antibiotic-resistance genes test, there are also blaCTX, blaKPC… etc. could be used. Suggestion for the author, the selection of these antibiotic-resistance genes would be based upon the results of the antimicrobial susceptibility testing, moreover, the selection of the current virulence genes based upon their signifcant role in the pathogenesis of the disease, that all on your test bacterial strains characteristic in this research. Basically, the phoA gene is considered to be as a marker to recognize secreted proteins, also critical to the virulence of enteropathogenic E. coli, but most of the E. coli in this study belong to ETEC and EHEC. Can't just only choose phoA gene used as marker. In a similar situation, the blaTEM gene that one of β-lactamase producers. It is not enough to only use the blaTEM for testing.

Response 5: rewritten (seen lines 267 to 270 and lines 283 to 290)

The alkaline phosphatase gene (phoA) is used in all E. coli strains and has previously been used in PCR-based methods for strain detection with high specificity (Stratakos, A. C., Linton, M., Millington, S., & Grant, I. R. (2017). A loop‐mediated isothermal amplification method for rapid direct detection and differentiation of nonpathogenic and verocytotoxigenic Escherichia coli in beef and bovine faeces. Journal of applied microbiology, 122(3), 817-828.)

As a result, the phoA gene was chosen as a target for identification of all E. coli strains in this analysis (both related and Non-related strains).

The blaTEM is common in non-O157 STECs in animals, and consequently, the high rates of ampicillin-resistant blaTEM-positive isolates were to be expected (Maidhof et al, 2002 and Kennedy et al, 2017).

The blaTEM was identified in non-O157 STEC isolated from cattle feces and soil samples, but none of the other blaOXA, blaSHV, and blaCTX-M variants could be detected (Scott et al, 2009). Poultry-associated STEC is more likely to carry the blaSHV variant,and blaOXA, blaSHV, and blaCTX-M have been previously reported in STEC O157:H7 Serotypes (Ahmed and  Shimamoto. 2015).

As a result, the blaTEM gene was chosen as a target for identification of non-O157 STEC strains isolated from beef carcasses and slaughterhouse settings.

Maidhof, H., B. Guerra, S. Abbas, H.M. Elsheikha, T.S. Whittam, and L. Beutin. 2002. A multiresistant clone of Shiga toxin-producing Escherichia coli O118:[H16] is spread in cattle and humans over different European countries. Appl. Environ. Microbiol. 68:5834–5842.

Kennedy, C. A., Fanning, S., Karczmarczyk, M., Byrne, B., Monaghan, Á., Bolton, D., & Sweeney, T. (2017). Characterizing the multidrug resistance of non-O157 Shiga toxin-producing Escherichia coli isolates from cattle farms and abattoirs. Microbial Drug Resistance23(6), 781-790.

Scott, L., P. McGee, C. Walsh, S. Fanning, T. Sweeney, J. Blanco, M. Karczmarczyk, B. Earley, N. Leonard, and J. Sheridan. 2009. Detection of numerous verotoxigenic E. coli serotypes, with multiple antibiotic resistance from cattle faeces and soil. Vet. Microbiol. 134:288–293

Ahmed, A.M., and T. Shimamoto. 2015. Molecular analysis of multidrug resistance in Shiga toxin-producing Escherichia coli O157: H7 isolated from meat and dairy products. Int. J. Food Microbiol. 193:68–73.

Point 6: Some data presentation methods will mislead readers. In this research tested the antimicrobial resistance of 20 E. coli isolate the results in stable 3, all tested isolates were 100% resistant to Penicillin. Readers may misunderstand that all E. coli isolates have resistance to Penicillin.

Response 6: Rewritten  (see lines 134 to 136 and lines  232 to 234)

Anti-microbial resistance (AMR) was determined against 14 antimicrobials, for 20 selective isolates from carcass shoulder, brisket and thigh and from abattoir effluent (5 from each representing the serotypes O166, O146, O44, O111 & O26). See lines 133 to 135

This research tested the antimicrobial resistance of 20 E. coli isolates, selected from carcass shoulder, brisket and thigh and from abattoir effluent (5 from each represent the serotypes O166, O146, O44, O111 & O26). See lines 215 to 217

Point 7: The research results in Table 1 and Table 2 show that the author has at least 110 different E. coli strains in this study. However, there are only 20 E. coli strains for antimicrobial susceptibility. The author needs to explain the relationship and differences between these 20 strains and the 110 strains. Why did the 110 strains not be tested for Antimicrobial susceptibility? How the 20 strains were selected and decided, and the representativeness of the entire study should also be explained clearly.

Response 7: Rewritten ( see lines 134 to 136 and lines

Anti-microbial resistance (AMR) was determined against 14 antimicrobials, for 20 selective isolates from carcass shoulder, brisket and thigh and from abattoir effluent (5 from each representing the serotypes O166, O146, O44, O111 & O26).

Furthermore, 20 isolates were chosen as a target for determination of Anti-microbial resistance by representing the sample location and the isolated serotypes

Point 8: In the experimental result analysis table in Table 3, these strains are independent and different, and the properties and conditions of the 14 antibiotics tested are also different. Therefore, the minimum requirement statistical analysis methods should be used to show the correlation or error of the result data.

Response 8: Rewritten (see line 159)

Spss program, version 26, was used to perform the Chi-square test (significance level: P < 0.05).

Table 3. Antimicrobial susceptibility of E. coli strains isolated (N= 20).

Antimicrobial agent

Sensitive

Intermediate

Resistant

Penicillin (P)

-

-

20 (100%)

Erythromycin (E)

-

2 (20%)

18 (80 %)

Oxytetracycline (T)

3 (15%)

2 (10%)

15 (75%)

Nalidixic acid (NA)

4 (20 %)

3 (15%)

13 (65%)

Ampicillin (AM)

-

8 (40 %)

12 (60%)

Sulphamethoxazol (SXT)

6 (30%)

3 (15%)

11 (55%)

Cephalotin (CN)

9 (45%)

2 (10%)

9 (45%)

Enrofloxacin (EN)

10 (50 %)

2 (10 %)

8 (40 %)

Oxacillin (OX)

12 (60%)

1 (5 %)

7 (35 %)

Neomycin (N)

14 (70%)

-

6 (30%)

Chloramphenicol (C)

16 (80 %)

-

4 (20%)

Kanamycin (K)

15(75 %)

2 (10%)

3 (15%)

Ciprofloxacin (CP)

16 (80%)

2 (10 %)

2 (10%)

Gentamicin (G)

19 (95%)

-

1 (5%)

P value

P < 0.0001

P < 0.0001

P < 0.0001

Chi-Square Tests

Value

df

Asymptotic Significance (2-sided)

Pearson Chi-Square

115.172a

22

.000

Likelihood Ratio

125.579

22

.000

Linear-by-Linear Association

65.085

1

.000

N of Valid Cases

233

a. 14 cells (38.9%) have expected count less than 5. The minimum expected count is .21.

Point 9:   At the same time, it is suggested that the author add two columns to Table 3, namely serotype and disease type, so that it can show the relationship between serotype and pathogenic E. coli to the antibiotic resistance selected in this study.

Response 9:

Thanks for your comment, the study aims to investigate the multidrug-resistance traits of E. coli isolated from beef carcasses and slaughterhouse environment along with prevalence, MALDI-TOF MS protein mass-spectra profiles & polymerase chain reaction assay. Therefore, the isolated E. coli serotypes were represented symmetrically along with sample location (20 selective isolates, 5 isolates from 4 different sample location). So, I suggest there is no need to name the serotypes and disease types as they have already been mentioned in detail in Table 2.

Point 10: In the results of antibiotic resistance pattern and MAR index, MAR and final average values have been generated, but the results of individual antibiotic response are different. Please explain how to benchmark and statistical methods of individual antibiotic test results, so that readers will be published in the future easy to understand.

Response 10:

Thank you for your comment. I double-checked the table 4 findings and found no differences between them and the results of individual antibiotic response. For example, 7 out of 20 isolates in table 3 (individual antibiotic resistance pattern) were resistant to Oxacillin (OX). In table 4 was the same, with one isolate resistant to the given antibiotic in the first, second, third, fourth, and sixth rows, and two isolates resistant in the fifth row, for a total of seven isolates resistant to Oxacillin (OX).

Table 4. Antibiotic resistance pattern and MAR index of E. coli isolated (N= 20).

Resistance pattern

Resistance profile

Number of isolates

Number of antibiotics

MAR

              i. 

P, E, T, NA, AM, SXT, CN, EN, OX, N, C, K, CP, G

1

14

1

            ii. 

P, E, T, NA, AM, SXT, CN, EN, OX, N, C, K, CP

1

13

0.92

           iii. 

P, E, T, NA, AM, SXT, CN, EN, OX, N, C, K

1

12

0.85

           iv. 

P, E, T, NA, AM, SXT, CN, EN, OX, N, C

1

11

0.78

            v. 

P, E, T, NA, AM, SXT, CN, EN, OX, N

2

10

0.714

           vi. 

P, E, T, NA, AM, SXT, CN, EN, OX

1

9

0.642

          vii. 

P, E, T, NA, AM, SXT, CN, EN

1

8

0.571

        viii. 

P, E, T, NA, AM, SXT, CN

1

7

0.5

           ix. 

P, E, T, NA, AM, SXT

2

6

0.428

            x. 

P, E, T, NA, AM

1

5

0.357

           xi. 

P, E, T, NA

1

4

0.285

          xii. 

P, E, T

2

3

0.21

        xiii. 

P, E

3

2

0.142

        xiv. 

P

2

1

0.071

Average                                                                                                                             0.533

Point 11:  The variations in the mass spectrum of E. coli isolates are showed with the representative example of typing dependent on MALDI-TOF MS. However, there are that the author did not explain and analyze the typing dependent data results shown.

Response 11: Rewritten (see lines 176 to 185)

The validity of peak patterns as taxonomic markers is set up utilizing MALDI-TOF MS as a bacterial fingerprinting method. findings showed that these criteria are met in the sample we tested in this analysis. MS profiles are reliable when a group of isolates is studied simultaneously under the same experimental and instrumental conditions, ensuring high repeatability and reproducibility across studies. Mass spectra, on the other hand, seem to have an inherent variability. All the tested isolates had mass peak profiles between 2,000 and 12,000 m/z ratio. To visually distinguish spectra, the five discriminatory regions were extended. The ability of MALDI-TOF MS to discriminate may be used to screen E. coli strains isolated from beef carcasses and slaughterhouses. This indicates a strong association with other methods of identification and stronger diagnostic tests.

Point 12: The author uses principal component analysis to strengthen the presentation of experimental results and reduce the influence of interference factors on the results. PC1 and PC2 showed the highest discrimination potential and showed a cluster of isolates, while the unrelated isolates were not very similar. Especially in Figure 2, 3 and Line 181-189. The related and unrelated isolates that the definition and difference of the related and unrelated isolates should be explained here, so that readers can understand their differences and importance. But it is a pity that the author did not explain, analyze and interpret the results shown. The author must explain, analyze and discuss these important experimental results, so that readers can understand the spirit of these research experiments and the value of research results.

Response 12: Rewritten (see lines 199 to 203)

Thank you for your comment

The results showed in figures 3 and 4 Explain the relationship and source of contamination between the isolated serotypes from beef shoulder (five serotypes: O166, O146, O44, O111 & O26) and isolated serotypes from knives (four serotypes: O166, O44, O111 & O26) and workers' hands (four serotypes: O166, O146, O44 & O111), which were all tested at the same time of sampling. Clusters were precisely delineated by a PCA-based dendrogram and virtual gel view (Figures 3 and 4), defining nine closely related isolates and four unrelated E. coli isolates. In less than one day, all results from MALDI-TOF MS, including the thorough analysis of peak frame changes, were obtained. Isolates representing related E. coli strains were unclear from one another but were delineated from non-related strains. In addition, figure 3 and 4 revealed highly distinguishable diverse peaks within non-related E. coli isolates.

Point 13:  For the isolate, the protein expression peak produced by MS can be seen. Irrelevant isolates are marked in red in Fig. 4. However, in the chapter on materials and methods, Virtual gel is missing. Please add the source of experimental materials, well/ Lane concentration, electrophoresis method, and analysis parameters of electrophoresis graphs, in order to complete the readers to understand its differences and comparison benchmarks.

Response 13: Rewritten (see lines 130 & 131)

The MALDI Biotyper 3 software was used to convert the achieved spectra into a virtual gel (pseudo-gel-like) format.

Point 14: The source of the 10 strains of coliform bacteria used in Figure 5 and Figure 6 has not been explained, and the significance of the two experimental results is not clear. The author is requested to provide a clear explanation and description. Line 246-277 has explained that the detected phoA and blaTEM gene in E. coli have different detection rates from different meat sources. At this moment, only 10 strains are shown in the 2 figures, and all the detected strains are shown. The author has fully and detailed explanation and analysis are necessary so as not to misunderstand readers.

Response 14: Rewritten (see lines 258 to 262)

Thank you for your comment

The presence of virulence, and antibiotic-resistance genes of E. coli isolated from beef carcass and abattoir environment samples was investigated using a PCR assay of the virulence-determinant gene (phoA) and the antibiotic-resistance genes blaTEM. For the PCR assay, serotypes isolated from beef shoulder and abattoir effluent (five from each: O166, O146, O44, O111, and O26) were chosen.

The significance of the two experimental results were illustrated in response 5

Point 15: This research is very commendable for analyzing and discussing the characteristics of meat, work equipment and environmental E. coli, but it failed to study and confirm the relationship between these strains collected from different sources. For example, an E. coli was detected separately in the intestines, meat, utensils, personnel, and environment/waste water during the manufacturing process, and it was confirmed that it was the transmission of the same strain. It is very pity that Line 292-295 failed to specifically address the policy formulation, decision-making and resource allocation of health care services. For the prevention of MDR occurrence in livestock and avoiding the spread of slaughter or food to humans, it failed to point out the priorities and control methods in Line 289-291.

Response 15: Rewritten (see lines 339 to 357)

Thank you for your comment

This research aimed to assess the prevalence, MALDI-TOF MS protein mass-spectra profiles, multidrug-resistance traits, polymerase chain reaction detection of virulence, and antibiotic-resistance genes of E. coli isolated from beef carcasses and slaughterhouse environment. The study was effective in identifying the prevalence of  MDR-E. coli and virulent non-O157 STEC serotypes in meat and the slaughterhouse settings, as well as the extent of its danger to public health and resistance to several antimicrobials that pose a threat to human health, as the spread of multidrug-resistant (MDR) bacteria has been repeatedly warned of. Anti-microbial resistance (AMR) was determined against 14 antimicrobials and Multiple antibiotic resistance (MAR) index was investigated. The MALDI-TOF MS technique, on the other hand, was used to identify certain species in a short amount of time, in tandem with a study of the relationship between the distribution of those species in various sources, as revealed by the results of Principal component analysis (PCA). The ability of MALDI-TOF MS to discriminate E. coli strains isolated from beef carcasses and slaughterhouses could be used to screen them. This points to a close link between other methods of identification and more powerful diagnostic tests. The presence of these genes in all isolated serotypes from beef shoulder and abattoir effluent samples was also investigated using a PCR assay of some selected genes in the local environment and in beef carcasses samples. These findings suggest that these isolates have the same origin, necessitating vigilance and the development of policies and strategies regarding antimicrobial use in food animals and rapid Screening for effective MDR-E. coli and virulent non-O157 STEC control in the Slaughterhouses and the application of approved hygienic procedures.

Best Regards

Mohamed Tharwat Elabbasy

Reviewer 2 Report

The manuscript is interesting and well structured with interesting results related to the diffusion of E coli in the beef carcasses and slaughterhouse environment. However, minor revisions:

Abstract: line 21, replace the comma with the full stop

Materials and methods: specify, because it is not clear, whether swabs or sponges were used for sampling

  Paragraph 2.2  Change the heading "Identification of bacterial strains" to isolation and identification, since the cultivation techniques are also reported

Clarify and specify which conventional methods are used for identification before proceeding with  MAldi-tof confirmation 

Table 1 list the samples in descending order of prevalence or in alphabetical order

Figure 4. Use a clear image with better resolution

Table 2 list the names in alphabetical order

Author Response

Response to Reviewer 2 Comments

Point 1: Extensive editing of English language and style required

Response 1:

English editing was done

The English editing was done by Co-author No. 7 "Adeniyi A Adeboye," a native speaker from the United Kingdom.

Point 2: Abstract: line 21, replace the comma with the full stop

Response 2: Done

Point 3: Materials and methods: specify, because it is not clear, whether swabs or sponges were used for sampling

Response 3: Done

It was clarified in the whole manuscript that the sampling done using sterile sponge, all swab words were removed (see line 24, 28, 99, 104, 108 &166)

Point 4:  Paragraph 2.2  Change the heading "Identification of bacterial strains" to isolation and identification, since the cultivation techniques are also reported

Response 4: Done, See line 110

Isolation and identification of bacterial strains

Point 5: Clarify and specify which conventional methods are used for identification before proceeding with  Maldi-tof confirmation 

Response 5: Rewritten (see lines 111 to 118)

Thank you for your comment

 Analyses of samples were carried out according to ISO/CEN 13136:2012 (ISO/TS 13136, 2012).  Each sponge was placed into a stomacher bag containing 500 ml of mod-ified trypticase soy broth containing 8 mg/L novobiocin. Each sponge was blended for two minutes and incubated for 20 hours at 41.5 °C.  One ml of each sample was cul-tured on MacConkey agar medium and Levine-eosin methylene blue agar (Himedia, Mumbai, India) and followed by incubation at 37 °C. Sorbitol fermentation was tested on sorbitol MacConkey agar and sorbitol phenol red agar media (Himedia, Mumbai, India) (overnight incubation at 37 °C).

Point 6: Table 1 list the samples in descending order of prevalence or in alphabetical order

Response 6 : Done

Table 1. Prevalence of E. coli in examined samples collected samples (n = 20).

Samples

Prevalence

Abattoir effluent

20 (100%)

Abattoir floor

20 (100%)

Abattoir wall

11 (55%)

Beef brisket

15 (75%)

Beef shoulder

16 (80 %)

Beef thigh

13 (65%)

Knives

9 (45%)

Water

Not detected

Worker hands

6 (30%)

Point 7: Figure 4. Use a clear image with better resolution

Response 7 : Done

Point 8: Table 2 list the names in alphabetical order

Response 8 : Done

Table 2. Serological identification of E. coli isolated.

O166

O146

O44

O111

O26

Abattoir effluent

3

2

4

8

3

Abattoir floor

3

4

5

6

2

Abattoir wall

2

3

2

2

2

Beef brisket

4

2

3

3

3

Beef shoulder

2

4

1

7

2

Beef thigh

3

1

2

4

3

Knives

3

-

2

3

1

Worker hands

1

2

1

2

-

Total

21(19.10%)

18(16.36%)

20(18.18%)

35(31.81%)

16(14.54%)

Types

ETEC

EPEC

EPEC

EHEC

EHEC

Best Regards

Mohamed Tharwat Elabbasy
